# Postural instability and gait disturbance are associated with abnormal stereopsis in Parkinson's disease

**Jungyeun Lee, Sung Hoon Kang, Seong-Beom Koh** *

Department of Neurology, Korea University Guro Hospital, Korea University College of Medicine, Seoul, Republic of Korea

* parkinson@korea.ac.kr

## Abstract

### Background

Visual dysfunction, including abnormal stereopsis, is a significant non-motor symptom in Parkinson's disease (PD) that can reduce quality of life and appears early in the disease. Abnormal stereopsis is associated with worsening of bradykinesia and freezing of gait, though the exact pathways linking stereopsis to motor symptoms remain unclear. Furthermore, in PD patients, the pedunculopontine nucleus and laterodorsal tegmental complex play an active role in sensorimotor control, and these areas provide cholinergic projections. Cholinergic degeneration may be associated with symptoms such as abnormal stereopsis, postural instability, gait disturbances and cognitive impairment. Therefore, in this study, we hypothesized that a high postural instability and gait disturbance score would increase the risk of abnormal stereopsis in PD.

### Methods

We designed a cross-sectional study and included 240 early PD patients without ophthalmologic problems other than abnormal stereopsis. To evaluate stereopsis, we used Titmus stereo test plates. Stereopsis testing was performed only once at the time of the patient's initial PD diagnosis. We collected data from medical history taking, scales, cognitive function tests, gait analysis, and tilt table tests. To analyze the potential risk factors for abnormal stereopsis in PD, we conducted a binary logistic stepwise selection analysis.

### Results

Among the total of 240 PD patients, 185 were in the normal stereopsis group and 55 were in the abnormal stereopsis group. In the analysis for risk factors related to abnormal stereopsis, the postural instability and gait difficulties (PIGD) subtype score was significantly associated with abnormal stereopsis. (95% confidence interval: 1.37–5.15).

**Data Availability Statement:** Data cannot be shared publicly because the dataset contains potentially identifying and sensitive patient information. This study was conducted using medical records of patients with Parkinson's

disease, and public sharing of the data would compromise patient privacy. Data are available from the Institutional Review Board of Korea University Guro Hospital (contact via parkinson@korea.ac.kr) for researchers who meet the criteria for access to confidential clinical data. Researchers interested in accessing the data must submit a formal application to the IRB committee and sign a data access agreement to ensure appropriate data protection and ethical usage.

**Funding:** The author(s) received no specific funding for this work.

**Competing interests:** The authors have declared that no competing interests exist.

## Conclusions

In PD, particularly in PIGD subtype patients, abnormal stereopsis can lead to a decrease in the quality of sensory information, potentially interfering with feedback and adaptation processes. This, in turn, can negatively affect posture and gait, creating a vicious cycle.

## Introduction

Visual dysfunction is among the various non-motor symptoms in Parkinson's disease (PD) [1–5]. Visual dysfunction in PD, which can be identified in the prodromal stage, can significantly reduce quality of life [4, 5]. One specific aspect of visual dysfunction in PD is abnormal stereopsis, a problem with the perception of depth created by the brain's processing of binocular vision. Notably, abnormal stereopsis can be observed relatively early in the disease's progression [6]. Also, patients with abnormal stereopsis have worse bradykinesia compared to those with normal stereopsis [7]. It has been observed that the group of PD with freezing of gait demonstrated worse stereoacuity compared to the group without this symptom [8]. However, researchers have not yet elucidated the specific pathways as to why abnormal stereopsis occurs in PD patients, or how it is related to motor symptoms.

The pedunculopontine nucleus (PPN) plays a crucial role in gait disturbance and posture in PD. It is also an area that affects freezing of gait, a disabling symptom uniquely found in PD [9]. This region consists of neurons projecting both ascending and descending between the cortex, thalamus, basal ganglia, cerebellum, and spinal cord, performing various functions [10]. Among these, PPN cholinergic projections have an extended effect on midbrain dopaminergic systems by innervating neurons in the substantia nigra pars compacta and ventral tegmental area [10]. By modulating dopamine release, it can also influence striatal input from the cortex and thalamus [10].

The neurotransmitter balance in peripheral organs, such as the retina, also affects stereopsis. The cholinergic system and dopaminergic deficits in the retina are involved in interocular interaction and mixed percepts by regulating the balance of excitatory and inhibitory signals in the binocular visual cortex [11]. Abnormal stereopsis can be affected by decreased visual acuity of one eye and decreased color perception [3, 7]. A dopamine deficit in the retina and other structures of the peripheral visual system can impair visual function, including stereopsis [12, 13]. According to previous research using functional magnetic resonance imaging, abnormalities in stereopsis in PD are linked to atrophy of the non-dominant extrastriate cortex [14]. Abnormal stereopsis is involved with various neural circuits including the thalamus and posterior parietal lobe [3]. When considering the findings from various previous studies, abnormal stereopsis is probably the result of various neurotransmitters acting in different areas.

This study was performed to analyze the differences in symptoms and various test results between PD patients with abnormal stereopsis and those without and to identify the factors that increase the risk of abnormal stereopsis. Specifically, to explore the relationship between posture, gait, and stereopsis, all of which involve PPN cholinergic projections, we hypothesized that a high postural instability and gait disturbance score would increase the risk of abnormal stereopsis in PD.

## Materials & methods

### Patients

This was a cross-sectional study conducted at the Korea University Guro Hospital, a tertiary center in Seoul, Korea, from January 1, 2016 to December 31, 2020. A total of 240 patients

newly diagnosed with PD during this period were included in this study. The inclusion criteria were individuals who were diagnosed with PD by an experienced movement disorders specialist according to the United Kingdom Parkinson's Disease Society Brain Bank criteria [15]. Patients diagnosed with atypical parkinsonism (multiple system atrophy, progressive supranuclear palsy, dementia with Lewy bodies, and corticobasal syndrome) were excluded. Patients with secondary parkinsonism (vascular parkinsonism and hydrocephalus) and tumor also excluded. Patients with strabismus, nystagmus, visual field defects and significant differences in visual acuity between both eyes (more than 20/40 on the Snellen chart) were also excluded from the study. Additionally, those presenting conditions such as marked cataract, corneal opacity, glaucoma or retinal disorders potentially affecting contrast sensitivity were not included.

Ethical approval was obtained from the Institutional Review Board of the Korea University Guro Hospital (IRB number: 2012GR0099). Written informed consent in Korean was obtained from all patients before their inclusion in the study.

### Data collection

Data were collected through reviewing electronic medical records between March 1, 2021 and March 31, 2021. The personal information was kept confidential, and each individual was assigned anonymously. All data were entered into the electronic medical record at the time of the patients' initial diagnosis of PD. Two neurologists collected the data from medical records including clinical information such as patient histories, disease duration, symptoms, neurological examination results, scale scores, neurophysiological test results, cognitive function test (components of the Seoul Neuropsychological Screening Battery, and the Mini-Mental State Examination) results and stereopsis test results. All tests and scores were conducted in the off state within one month of the patient's diagnosis. Tremor dominant (TD) and postural instability and gait difficulty (PIGD) subtypes of PD were defined using the Unified Parkinson's Disease Rating Scale (UPDRS). The TD score was derived from UPDRS items 16, 20, and 21; the PIGD score was derived from UPDRS items 13–15 and 29–30. The ratio of the mean TD score to the mean PIGD score was used to define the TD subtype ($\geq 1.5$), the PIGD subtype ($\leq 1$), and the indeterminate subtype ($> 1$ and $< 1.5$) [16].

### Tests for abnormal stereopsis

Stereopsis was assessed using Titmus stereo test plates on the same day as the gait analysis and UPDRS assessment (Stereo Optical Co. Inc., Chicago, Illinois, USA). Normal stereopsis was characterized as having an arc measurement of 60 seconds or less in the Titmus fly test [6, 14, 17].

### Tests for gait parameters

A gait analyzer (GaitRite. CIR system, Inc. - 12 fit AP1105; CIR systems, Franklin, NJ, USA) was used to quantify gait. All patients walked ten times at their usual pace on a mat with embedded sensors. In each cycle, the spatial and temporal parameters of gait were recorded. Ten recordings were integrated.

### Tests for tilt table test

All patients refrained from taking medications and drinks that could influence autonomic function, such as coffee and alcohol, for 24 hours prior to the test. Blood pressure and heart rate were intermittently recorded at 1-minute intervals using an automated sphygmomanometer on the brachial artery, along with electrocardiogram monitoring. Initially, patients were

placed in a supine, head-down position on the tilt table for 10 minutes, followed by a transition to an 80˚ head-up tilt (standing) position for another 10 minutes. Orthostatic hypotension was defined as a sustained decrease of 20 mmHg in systolic blood pressure and/or 10 mmHg in diastolic blood pressure within 3 minutes of assuming the 80˚ head-up tilt position [18, 19].

### Outcomes, covariates, missing data

Our primary outcome was the presence of abnormal stereopsis in patients with PD. The secondary outcome was subjective non-motor symptoms, and the covariates included age, sex, hypertension, diabetes, duration of symptoms, presence of orthostatic hypotension, and results of cognitive function tests related to vision. Because only patients who completed all tests were included, there was no missing data.

### Statistical analysis

For the comparison of baseline characteristics, the Shapiro-Wilk normality test was conducted for continuous variables to assess normality. If the data followed a normal distribution, the Student's t-test was applied; otherwise, the Mann-Whitney U test was used. For categorical variables, the Chi-square test or Fisher's exact test was used. To analyze the associations between PIGD score and GaitRite parameters, Spearman's correlation test was used. To analyze the associations between potential risk factors and abnormal stereopsis in PD patients, we conducted a stepwise regression analysis (using step function in R, backward elimination and forward selection models). $p$-values $< 0.05$ were considered significant. The eliminated variables were: age, sex, education years, hypertension, diabetes, presence of orthostatic hypotension, Hoehn and Yahr (HY) stage, UPDRS part II, UPDRS part III, abnormal Mini-Mental State Examination, abnormal digit span backward, abnormal Rey-Osterrieth Complex Figure Test (copy, delayed recall), abnormal Controlled Oral Word Association test (animal, phonemic), abnormal Stroop Color Reading test, tremor score, and Non-Motor Symptom Scale total score. The variables selected in the regression model were UPDRS part I, abnormal Seoul Verbal Learning test, and PIGD score. All statistical analyses were performed using R software version 4.2.3 (https://www.r-project.org).

### Results

A total of 240 patients were included in the analysis of factors associated with abnormal stereopsis in PD. Among these 240, 185 patients were in the normal stereopsis group and 55 patients were in the abnormal stereopsis group. The mean age of all patients was 69.8 years, and 126 patients (52.5%) were female. Median disease duration was 12 months (Interquartile range: 7–36 months). Table 1 shows the differences in demographics and characteristics between the two groups. There were no significant differences in age, sex, and years of education. However, the Hoehn and Yahr scores showed a significant difference ($p$ = 0.024). There were no significant differences in PD subtype or UPDRS part III scores ($p$ = 0.529). The z-scores of the Rey-Osterrieth Complex Figure Test also showed a significant difference between two groups ($p$ = 0.032).

Table 2 presents the subscales of UPDRS part III between the two groups. No significant differences were detected at the subscale level. However, analysis of individual items revealed significant differences in 'arising from chair', 'gait', and 'postural instability'.

S1 Table shows the analysis of gait parameters between the two groups. Mean velocity was 85.41 and 83.77 centimeters per second, respectively ($p$ = 0.602). Mean cadence was 106.71 and 107.22 steps per minute, respectively ($p$ = 0.714). Differences in all parameters were insignificant.

**Table 1. Demographic and clinical characteristics of enrolled patients.**

| | PD_NL | PD_AS | Total | *p*- value |
|---|---|---|---|---|
| | (n = 185) | (n = 55) | (n = 240) | |
| Demographics | | | | |
| Age, y | 69.7 [68.57–70.83] | 70.0 [68.01–72.06] | 69.8 [68.80–70.76] | 0.779 |
| Sex, female | 99 (53.5%) | 27 (49.1%) | 126 (52.5%) | 0.672 |
| Education, y | 9.0 [6.0; 12.0] | 9.0 [6.0; 12.0] | 9.0 [6.0; 12.0] | 0.787 |
| Hypertension | 83 (44.9%) | 21 (38.2%) | 104 (43.3%) | 0.470 |
| Diabetes | 36 (19.5%) | 11 (20.0%) | 47 (19.6%) | 1.000 |
| Clinical characteristics | | | | |
| Disease duration, m | 12.0 [7.0; 30.0] | 18.0 [8.0; 36.0] | 12.0 [7.0; 36.0] | 0.347 |
| **H&Y stage** | **2.0 [2.0; 2.0]** | **2.0 [2.0; 2.5]** | **2.0 [2.0; 2.5]** | **0.024** |
| UPDRS part I | 2.0 [1.0; 3.0] | 2.0 [1.0; 3.0] | 2.0 [1.0; 3.0] | 0.547 |
| UPDRS part II | 8.0 [5.0; 9.0] | 8.0 [6.0; 11.5] | 8.0 [5.0; 10.0] | 0.227 |
| UPDRS part III | 23.0 [17.0; 29.0] | 24.0 [18.0; 28.0] | 23.0 [17.5; 29.0] | 0.529 |
| Subtype | | | | 0.963 |
| Tremor dominant | 56 (30.3%) | 17 (30.9%) | 73 (30.4%) | |
| PIGD | 101 (54.6%) | 29 (52.7%) | 130 (54.2%) | |
| Intermediate | 28 (15.1%) | 9 (16.4%) | 37 (15.4%) | |
| NMSS domain 1: Cardiovascular including falls | 0.0 [0.0; 1.0] | 0.0 [0.0; 1.0] | 0.0 [0.0; 1.0] | 0.226 |
| NMSS domain 4: Perceptual problems/hallucinations | 0.0 [0.0; 0.0] | 0.0 [0.0; 0.0] | 0.0 [0.0; 0.0] | 0.592 |
| NMSS domain 6: Gastrointestinal tract | 1.0 [0.0; 3.0] | 1.0 [0.0; 3.5] | 1.0 [0.0; 3.0] | 0.786 |
| NMSS domain 7: urinary | 5.0 [1.0; 7.0] | 5.0 [2.5; 9.0] | 5.0 [1.0; 8.0] | 0.283 |
| NMSS total score | 27.0 [18.0; 38.0] | 27.0 [13.5; 39.0] | 27.0 [17.0; 38.0] | 0.806 |
| Tilt table test (Orthostatic hypotension %) | 57 (30.8%) | 20 (36.4%) | 77 (32.1%) | 0.542 |
| MMSE (abnormal %) | 39 (21.1%) | 15 (27.3%) | 54 (22.5%) | 0.434 |
| Digit Span Test backward (z-score) | -0.2 [-0.8; 0.4] | -0.3 [-0.9; 0.6] | -0.2 [-0.9; 0.4] | 0.982 |
| **RCFT copy (z-score)** | **0.3 [-0.6; 0.8]** | **-0.3 [-1.1; 0.6]** | **0.2 [-0.7; 0.8]** | **0.032** |
| SVLT delayed recall (z-score) | -0.6 [-0.75 –-0.43] | -0.7 [-0.99 –-0.37] | -0.6 [-0.75 –-0.47] | 0.589 |
| RCFT delayed recall (z-score) | -0.5 [-1.2; 0.4] | -0.6 [-1.1; 0.3] | -0.5 [-1.1; 0.4] | 0.685 |
| COWAT animal (z-score) | -0.3 [-0.41 –-0.1] | -0.3 [-0.59 –-0.07] | -0.3 [-0.41 –-0.14] | 0.645 |
| COWAT phonemic (z-score) | -0.4 [-1.0; 0.3] | -0.5 [-1.2; 0.1] | -0.4 [-1.0; 0.2] | 0.249 |
| Stroop Color Reading (z-score) | -0.2 [-1.1; 0.7] | -0.6 [-1.7; 0.5] | -0.3 [-1.3; 0.7] | 0.101 |

Values are presented as mean [confidence interval], median [interquartile range] or number (%). The *P*-value was obtained using the Student's t-test if the data followed a normal distribution; otherwise, the Mann-Whitney U test was used. For categorical variables, the Chi-square test or Fisher's exact test was used. Abbreviations: PD_NL = Parkinson's disease with normal stereopsis; PD_AS = Parkinson's disease with abnormal stereopsis; UPDRS = Unified Parkinson's Disease Rating Scale; GI = gastrointestinal; NMSS = Non-Motor Symptoms Scale; MMSE = Mini Mental State Examination; RCFT = Rey-Osterrieth Complex Figure Test; SVLT = Seoul Verbal Learning Test; COWAT = Controlled Oral Word Association Test.

Table 3 shows the correlation between PIGD score and gait analysis parameters. Most parameters were correlated with PIGD score, except for cadence, step length differential, both step times, left swing time, and right single support time.

Table 4 shows the risk factor of abnormal stereopsis in PD. In the association analysis, a high PIGD subtype score (odds ratio 2.65, *p*-value 0.004) was associated with abnormal stereopsis after adjusting for UPDRS part I score and abnormal SVLT result.

## Discussion

We aimed to investigate the differences in symptoms and various test results between PD patients with abnormal stereopsis and those without. We hypothesized that a higher PIGD

**Table 2. The subscales of UPDRS part III in enrolled patients.**

|  | PD_NL | PD_AS | *p*- value |
|---|---|---|---|
|  | (n = 185) | (n = 55) |  |
| Subscale of axial and gait | 5.00 [4.00, 6.00] | 6.00 [4.00, 8.00] | 0.063 |
| Subscale of resting tremor | 1.00 [0.00, 3.00] | 1.00 [0.00, 3.00] | 0.892 |
| Subscale of postural tremor | 1.00 [0.00, 2.00] | 1.00 [0.00, 2.00] | 0.311 |
| Subscale of tremor | 2.00 [1.00, 4.00] | 2.00 [0.50, 5.00] | 0.601 |
| Subscale of rigidity | 4.00 [2.00, 6.00] | 4.00 [2.00, 6.00] | 0.702 |
| Subscale of bradykinesia left | 6.00 [4.00, 8.00] | 6.00 [4.50, 7.00] | 0.804 |
| Subscale of bradykinesia right | 5.00 [3.00, 7.00] | 5.00 [4.00, 7.00] | 0.350 |
| Subscale of bradykinesia both | 10.81 [10.17–11.44] | 11.18 [9.92–12.44] | 0.380 |
| **UPDRS III sum** | 23.00 [17.00, 29.00] | 24.00 [18.00, 28.00] | 0.528 |

Values are presented as mean [Confidence interval] or median [Interquatile range]. The P-value was obtained using the Student's t-test if the data followed a normal distribution; otherwise, the Mann-Whitney U test was used. Abbreviations: PD_NL = Parkinson's disease with normal stereopsis; PD_AS = Parkinson's disease with abnormal stereopsis; UPDRS = Unified Parkinson's Disease Rating Scale.

score increases the risk of abnormal stereopsis. Consistent with previous studies, the results of our study indicate that PD motor symptoms are associated with the risk of stereopsis abnormalities [7, 8]. Specifically, in our study, higher PIGD scores increased the risk of stereopsis abnormalities, but bradykinesia was not associated with abnormal stereopsis, unlike previous research findings [7].

**Table 3. The correlations between the PIGD score and parameters of gait analysis.**

|  | Correlations with PIGD score | |
|---|---|---|
|  | **Correlation coefficient (rho)** | **P-value** |
| Velocity (cm/s) | **-0.455** | **<0.001** |
| Cadence (steps/min) | -0.108 | 0.094 |
| Step length differential (cm) | 0.117 | 0.071 |
| Left step time (seconds) | 0.103 | 0.112 |
| Right step time (seconds) | 0.071 | 0.277 |
| Left swing time (seconds) | -0.092 | 0.154 |
| Right swing time (seconds) | **-0.139** | **0.032** |
| Left stance time (seconds) | **0.200** | **0.002** |
| Right stance time (seconds) | **0.221** | **0.001** |
| Left single support time (seconds) | **-0.162** | **0.012** |
| Right single support time (seconds) | -0.116 | 0.074 |
| Left double support time (seconds) | **0.339** | **<0.001** |
| Right double support time (seconds) | **0.321** | **<0.001** |
| Left step length (cm) | **-0.480** | **<0.001** |
| Right step length (cm) | **-0.470** | **<0.001** |
| Left stride length (cm) | **-0.485** | **<0.001** |
| Right stride length (cm) | **-0.482** | **<0.001** |
| Left CV of step length | **0.354** | **<0.001** |
| Right CV of step length | **0.411** | **<0.001** |
| Left CV of stride length | **0.374** | **<0.001** |
| Right CV of stride length | **0.409** | **<0.001** |

Spearman analysis was performed. CV = Coefficient of Variation.

**Table 4. Factors associated with abnormal stereopsis in Parkinson's disease.**

| Variables entered | Odds ratio | 95% CI | p-value |
|---|---|---|---|
| UPDRS part I sum | 0.83 | 0.67, 1.02 | 0.074 |
| Abnormal SVLT | 1.84 | 0.98, 3.49 | 0.060 |
| PIGD score | 2.65 | 1.37, 5.15 | **0.004** |

A stepwise regression analysis (using step function in R, backward elimination and forward selection models). *p*-values < 0.05 were considered significant. Variables eliminated from the model: age, sex, education years, hypertension, diabetes, presence of orthostatic hypotension, Hoehn and Yahr stage, UPDRS part II, UPDRS part III, abnormal Mini-Mental State Examination, abnormal digit span backward, abnormal Rey-Osterrieth Complex Figure Test (copy, delayed recall), abnormal Controlled Oral Word Association test (animal, phonemic), abnormal Stroop Color Reading test, tremor score, and Non-Motor Symptom Scale total score.

Physiologically, cholinergic denervation is a major cause of PIGD features in PD [20]. The PPN, which plays a key role in posture and gait stability, is part of the PPN-laterodorsal tegmental complex and sends cholinergic projections to the thalamus [20]. This area is known to play an active role in sensorimotor control [21, 22]. There is a possibility that these projections are associated with abnormal stereopsis. Also, previous study has indicated that the cholinergic system is involved in stereopsis abnormalities [11]. The cholinergic system regulates the excitatory and inhibitory balance in the binocular visual cortex, and cholinergic enhancement in the visual cortex (V1) decreases inhibitory drive. This increases interocular interaction and mixed percepts [11]. Therefore, cholinergic denervation in PD may not only increase the risk of postural instability and gait disturbance but also increase the risk of stereopsis abnormalities. Additionally, abnormal stereopsis may lead to a qualitative degradation of visual information, potentially interfering with feedback and adaptation processes. This can further create a vicious cycle by negatively impacting posture and gait.

PIGD subtype patients are at a higher risk of cognitive decline [20, 23]. Abnormal stereopsis increases the risk of dementia conversion in PD [24, 25]. As shown in Table 1, the Rey complex figure copy score differs between the two groups, suggesting a possible association with cognition, consistent with the marked cognitive decline observed in PIGD patients [26, 27]. However, the possibility that abnormal stereopsis may have influenced the Rey Complex Figure Copy Test should also be considered. In logistic regression analysis, no cognitive function tests increased the risk of stereopsis abnormality. Nevertheless, the abnormal Seoul verbal learning test was selected in the model with a p-value of 0.06. Although this was not significant, it could potentially show meaningful results under different conditions, such as with a larger sample size.

Researchers have studied the relationship between visual acuity and the prognosis of PD, and some have suggested that loss of visual acuity is a symptom in the prodromal stage [4, 5, 28]. Although our results showed that patients with abnormal stereopsis had higher HY scores, HY was not selected as a risk factor in the regression analysis. Also, there was no significant difference in disease duration between the two groups. Median disease duration was 12 months, indicating that they can be considered early-stage patients. However, further research is needed to determine whether stereopsis worsens in advanced patients and whether it could serve as a marker of disease progression.

Visual acuity and stereopsis are closely related [29]. Stereopsis tests, such as the Titmus fly test, are relatively simpler and more cost effective than visual acuity tests. Conducting stereopsis tests during the prodromal stage could be useful and may help predict the prognosis of patients.

The study has several limitations. First, its retrospective study design and its being conducted at a single tertiary hospital are limitations. Additionally, it is possible that the enrolled patients constituted a unique cohort. We specified that the tests were conducted within one month of clinical diagnosis to minimize the influence of medication and maintain uniformity in the early stages of the disease. However, this approach limits generalization to the entire PD population, and this limitation needs to be further explored through additional research in future studies. Second, additional gait analysis using GaitRite did not reveal any differences between the two groups. Recent studies have demonstrated that combining mat with wearable devices or 3D cameras enables the detection of subtle movements in early-stage patients, which can be valuable for diagnostic purposes [30, 31]. However, the equipment in our hospital, which consisted of only a mat and an analytical computer, was limited in its ability to assess certain kinetics and kinematics parameters such as upper extremity motion and center of pressure. It is possible that differences not captured by this equipment may be reflected in the UPDRS scores, which are based on human assessment. It is not impossible for spatiotemporal parameters to detect gait abnormalities in early PD; however, the parameters we measured in our gait analysis may have been insufficient to establish a connection with the risk of abnormal stereopsis. Thus, further studies utilizing gait analysis with integrated upper extremity movement assessment are necessary to validate these assumptions. Third, no other ophthalmologic examinations were performed except for stereopsis. However, we tried to compensate for this limitation by excluding patients with ophthalmologic conditions through careful history-taking.

## Conclusion

In the early PD patients, especially in those with the PIGD subtype, abnormal stereopsis can lead to a decrease in the quality of sensory information, potentially interfering with feedback and adaptation processes. This, in turn, can negatively affect posture and gait, creating a vicious cycle. By identifying possible risk factors that increase stereopsis abnormalities at a relatively early stage, this study will help clinicians to predict patient prognosis and will also provide valuable information for future research.

## Supporting information

**S1 Table. Analysis of gait parameters across groups with normal and abnormal stereopsis.** (DOCX)

## Author Contributions

**Data curation:** Jungyeun Lee, Sung Hoon Kang.

**Formal analysis:** Jungyeun Lee.

**Investigation:** Jungyeun Lee, Sung Hoon Kang.

**Methodology:** Jungyeun Lee.

**Project administration:** Seong-Beom Koh.

**Supervision:** Seong-Beom Koh.

**Writing – original draft:** Jungyeun Lee.

**Writing – review & editing:** Sung Hoon Kang, Seong-Beom Koh.

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
