## [Decision Letter · Decision Letter 0]

8 Sep 2024

PONE-D-24-29202Postural instability and gait disturbance are associated with abnormal stereopsis in Parkinson's disease: A retrospective study of 240 patientsPLOS ONE

Dear Dr. Koh,

Thank you for submitting your manuscript to PLOS ONE. After careful consideration, we feel that it has merit but does not fully meet PLOS ONE’s publication criteria as it currently stands. Therefore, we invite you to submit a revised version of the manuscript that addresses the points raised during the review process.

We look forward to receiving your revised manuscript.

Kind regards,

Keisuke Suzuki, MD, PhD

Academic Editor

PLOS ONE

Journal Requirements:

When submitting your revision, we need you to address these additional requirements. 1. Please ensure that your manuscript meets PLOS ONE's style requirements, including those for file naming. The PLOS ONE style templates can be found at https://journals.plos.org/plosone/s/file?id=wjVg/PLOSOne_formatting_sample_main_body.pdf and https://journals.plos.org/plosone/s/file?id=ba62/PLOSOne_formatting_sample_title_authors_affiliations.pdf 2. Please provide additional details regarding participant consent. In the ethics statement in the Methods and online submission information, please ensure that you have specified (1) whether consent was informed and (2) what type you obtained (for instance, written or verbal, and if verbal, how it was documented and witnessed). If your study included minors, state whether you obtained consent from parents or guardians. If the need for consent was waived by the ethics committee, please include this information. If you are reporting a retrospective study of medical records or archived samples, please ensure that you have discussed whether all data were fully anonymized before you accessed them and/or whether the IRB or ethics committee waived the requirement for informed consent. If patients provided informed written consent to have data from their medical records used in research, please include this information. 3. We note that your Data Availability Statement is currently as follows: All relevant data are within the manuscript and its Supporting Information files. Please confirm at this time whether or not your submission contains all raw data required to replicate the results of your study. Authors must share the “minimal data set” for their submission. PLOS defines the minimal data set to consist of the data required to replicate all study findings reported in the article, as well as related metadata and methods (https://journals.plos.org/plosone/s/data-availability#loc-minimal-data-set-definition). For example, authors should submit the following data: - The values behind the means, standard deviations and other measures reported;- The values used to build graphs;- The points extracted from images for analysis. Authors do not need to submit their entire data set if only a portion of the data was used in the reported study. If your submission does not contain these data, please either upload them as Supporting Information files or deposit them to a stable, public repository and provide us with the relevant URLs, DOIs, or accession numbers. For a list of recommended repositories, please see https://journals.plos.org/plosone/s/recommended-repositories. If there are ethical or legal restrictions on sharing a de-identified data set, please explain them in detail (e.g., data contain potentially sensitive information, data are owned by a third-party organization, etc.) and who has imposed them (e.g., an ethics committee). Please also provide contact information for a data access committee, ethics committee, or other institutional body to which data requests may be sent. If data are owned by a third party, please indicate how others may request data access.

Reviewers' comments:

Reviewer's Responses to Questions

**Comments to the Author**

1. Is the manuscript technically sound, and do the data support the conclusions?

Reviewer #1: Yes

Reviewer #2: Partly

Reviewer #3: Yes

2. Has the statistical analysis been performed appropriately and rigorously? 

Reviewer #1: Yes

Reviewer #2: Yes

Reviewer #3: Yes

3. Have the authors made all data underlying the findings in their manuscript fully available?

Reviewer #1: Yes

Reviewer #2: Yes

Reviewer #3: Yes

4. Is the manuscript presented in an intelligible fashion and written in standard English?

Reviewer #1: Yes

Reviewer #2: Yes

Reviewer #3: No

5. Review Comments to the Author

Reviewer #1: This study is an interesting study with a well-designed structure. Still visual symptoms in PD are under-estimated. Especially, the authors recruited quite big number of drug-naive PD patients in this study, so they could exclude possible confounders, mainly seen in the advanced stage.

1. As mentioned before, enrollment of drug-naive PD patients can be a big strength of this study. However, the authors did not describe why they recruited drug-naive PD patients in the introduction section, so I recommend emphasizing the reason why they included drug-naive PD patients in introduction section and maybe in discussion section.

2. Some of the enrolled subjects may be diagnosed as atypical parkinsonism later. How long did the authors follow up after the diagnosis of PD? In particular, freezing of gait (FOG) is very rare in drug-naive PD patients, but can be seen in patients with atypical parkinsonism even from the early stage. In this study 7% of enrolled subjects had FOG, and it is better to check the data again.

3. The authors checked various symptoms, including constipation, anosmia, orthostatic dizziness, urinary incontinence, visual hallucination, easy falling and FOG. How did they define these symptoms? Any scale used to screen these symptoms (e.g. NMSS sub-items)?

4. In terms of table 2, how about using sub-score of UPDRS part 3 (e.g. sub-score for tremor, rigidity, bradykinesia and axial symptoms)? This would be more easy for the readers.

5. Results without statistical significance like the comparison of GAITRITE data can be moved to supplement.

6. Did the authors check RBD or dysautonomia (not just symptoms but autonomic function test or orthostatic hypotension)? Considering these symptoms can be markers to differentiate brain-first vs. body-first subtype, analysis with these variables can be also interesting.

Reviewer #2: This study is original and relevant in exploring the relationship between postural instability, gait disturbances, and stereopsis abnormalities in patients with Parkinson's disease. This topic has been little investigated to date. The results presented are interesting, suggesting a possible association between the PIGD subtype and visual dysfunction, which may open new perspectives on the impact of visual abnormalities on the disease's motor symptoms. However, despite the significant contribution, some aspects could be improved to strengthen the conclusions and increase the robustness of the findings. The most critical aspect is the absence of significant differences in the objective gait parameters between the groups. Can the relationship between gait/postural alterations and stereopsis be sustained based on scores in some items of UPDRS-section III?

Introduction: The introduction situates the study's relevance in the context of PD, mentioning stereopsis as an important non-motor symptom. However, the article could delve deeper into how visual dysfunction, specifically stereopsis, contributes to motor disturbances in PD. This more comprehensive exploration is crucial for providing a solid justification for the study and understanding the full impact of visual abnormalities on the disease's motor symptoms.

Methods: The adopted test to detect stereopsis is appropriate, but the authors do not mention its sensitivity and specificity in patients with PD. Please consider including this information to improve the interpretation of the results.

Results and Discussion: The results show differences between the groups regarding postural instability and gait according to UPDRS section III. However, the objective gait analysis did not show significant differences between the groups, even in parameters closely related to postural and gait alterations such as single and double support time and CV of step and stride length. Are the scores in UPDRS items used to classify the PIGD subtype correlated with these gait parameters? In the absence of differences in objective gait measures, further explanation is needed for the association between PIGD and abnormal stereopsis.

The study found a significant link between the PIGD subtype and abnormal stereopsis. However, based on postural control theories, the discussion regarding the connection between the PIGD subtype and stereopsis abnormalities must be deeper. The authors also suggested that cholinergic denervation in PD may be associated with abnormal stereopsis, but the explanation is unclear and does not effectively connect the study results to this theory.

Study limitations: Please make sure to include the unique cohort as one of the limitations to consider.

Reviewer #3: Title:

• The title is informative; however, it is long. The authors should consider shortening the title, and avoiding using the term retrospective because it is no longer recommended. Besides, this is a cross-sectional study, no matter if it uses data from a databank or if the authors seek the data prospectively.

Abstract:

• The text should be shortened without losing information. The authors should insert the setting and the study design.

• The main and secondary outcomes should be clearly stated.

• It is unclear what data was retrieved from the patient record. Were the patients examined routinely for the stereopsis diagnosis? In which moment? This must be clarified.

• The results lack measures of association from the logistic regression – please state also the confidence interval.

Introduction

• The text offers the reader a sufficient amount of information about stereopsis. Nevertheless, the flow of ideas is confusing across the paragraphs. Please consider combining the first two paragraphs to provide essential information on the prevalence of stereopsis in PD and its associations with other symptoms already defined in the literature.

• The second paragraph could be better organized when considering the pathophysiology of stereopsis in PD. Suggestion: begin with the neurochemical dysfunctions associated with stereopsis and finish with the findings in functional MRI.

• Finally, state the study's specific objectives, including any prespecified hypotheses based on the motivations for this research.

Methods

• Line 109: remove the term “retrospective”.

• Line 116: Patients with secondary parkinsonism (vascular parkinsonism, which presents as a severe white matter hyperintensity burden in magnetic resonance imaging and hydrocephalus): the information regarding MRI findings in vascular parkinsonism is irrelevant and excessive in the context of the study. The authors should consider removing it.

• Line 141: there is a typographic error in the phrase, please review.

• The patients underwent visual and gait analysis. Was this done at the same time, or at least with a small interval? What was the interval between the analyses, considering the impact of stereopsis on gait performance? This is crucial information regarding potential measurement biases.

• Line 153: the term "statistically significant" in the text (p-values < 0.05 were considered significant.) is no longer recommended since all p-values come from a statistical analysis, which is redundant. Please remove the term "statistically."

• The study design has been defined; however, it lacks statements about primary and secondary outcomes, covariates considered as possible confounders, and how the authors addressed missing data.

Results

• Line 163: The Hoehn & Yahr scale is ordinal and should not be summarized using means and SD, but rather medians and interquartile. Additionally, the statistical test should be suitable for ordinal data.

• Table 1 contains not only demographic data but also a statistical analysis comparing the groups, so the table titles should be adjusted.

• Table 1 also lacks the statistical tests used in each analysis.

• Table 1: The analyzed data should include the confidence interval of the mean in both groups, which is essential for the study's transparency and internal validity.

• Table 2: The analyzed data should include the confidence interval of the mean in both groups, which is essential for the study's transparency and internal validity. Besides, the statistical test used must be stated in the legend.

• Line 186: Differences in all parameters were statistically insignificant. The same observation for the term “statistically”.

• Table 3: the same recommendations for table 2 and 3 regarding the 95% CI and statistical test used.

• Lines 191-5: In the text (Table 4 shows the risk factor of abnormal stereopsis in PD. In the association analysis, a high PIGD subtype score (odds ratio 2.65, p-value 0.004) was associated with abnormal stereopsis after adjusting for UPDRS I score and abnormal SVLT result.), it is unclear which measure is significantly associated with stereopsis – was the binary definition of PIGD subtype (yes or not) or the numeric value of the PIDG score?

• The authors performed a logistic regression with stepwise modeling. What covariates were risk factors at the end? Why at the final exposition the authors state that PIGD score was adjusted for the UPDRS I score and abnormal SVLT result? Ideally, all the odds ratios should be presented. As a suggestion, a forest plot could be included containing all the variables used in the regression analysis and respective odds ratios.

• The authors must discuss the study biases and state if there were missing data.

Discussion

Overall, the text thoroughly discusses the research findings, but it lacks flow. The authors jump from one issue to another, confusing the reader.

• Line 231: “Table 1, the Rey complex figure copy score differs between the two groups, suggesting a possible association with cognition…”. The authors should consider the nature of the Rey Complex figure: this cognitive test is highly dependent on visual skills, and the stereopsis may have biased the results. This must be discussed. Considering the short mean duration of the disease (9 months) in this sample, it is not expected to have many demented patients.

• Line 235: “PIGD subtype may worsen stereopsis abnormalities and then cognitive function.”- PIGD subtype is a classification for PD subtypes (highly controversial nowadays).

• Being a PIGD subtype patient means that he or she has a higher burden of postural instability and gait disorders compared to tremor-dominant patients. This subtype is also associated with a higher decline in cholinergic function from the Meynert nucleus and PPN, which is linked to cognitive decline and gait disorders. Therefore, it's not the PIGD subtype itself that worsens these symptoms, but rather the neurotransmitter dysfunction associated with this subtype. Stereopsis may be one of the consequences of this dysfunction, linked with dopaminergic denervation in several anatomical points. The authors should revise the statement and discuss this appropriately.

• Abnormal stereopsis may be an early symptom in PD. However, in this sample, the group with abnormal stereopsis had a higher HY score. Despite the fact that HY was not a risk factor for abnormal stereopsis, the authors should discuss it in light of the literature findings.

The whole text should have an typographic and grammar edition.

6. PLOS authors have the option to publish the peer review history of their article (what does this mean?). If published, this will include your full peer review and any attached files.

Reviewer #1: No

Reviewer #2: **Yes: **Maria Elisa Pimentel Piemonte

Reviewer #3: No

---

## [Author Response · Author response to Decision Letter 0]

29 Oct 2024

Response to Reviewer 1

Comments to the Author

This study is an interesting study with a well-designed structure. Still visual symptoms in PD are under-estimated. Especially, the authors recruited quite big number of drug-naive PD patients in this study, so they could exclude possible confounders, mainly seen in the advanced stage.

Thank you for your review of our paper. We have answered each of your points below.

Comment 1: As mentioned before, enrollment of drug-naive PD patients can be a big strength of this study. However, the authors did not describe why they recruited drug-naive PD patients in the introduction section, so I recommend emphasizing the reason why they included drug-naive PD patients in introduction section and maybe in discussion section. ¬

Response 1: We greatly appreciate the insightful comment provided by the reviewer. We first apologize for the typo "de novo" in the abstract. This study enrolled patients who were newly diagnosed with Parkinson's disease at our hospital, and it remains uncertain whether these patients had taken anti-parkinsonian medications. The patients included in this study consist of a combination of drug-naive individuals and those who started medication within the past month. Therefore, we thought it necessary to revise “de novo” to “early.” According to the prior study (Sun et al., 2014), there was no significant difference in stereopsis between the on and off states. Nevertheless, to minimize the potential influence of medication, if the patients were on medication, we discontinued their medications during the initial work-up, including stereopsis tests and UPDRS assessments, and conducted the tests in the off state.

Abstract

"Methods: We designed a cross-sectional study and included 240 early PD patients without ophthalmologic problems other than abnormal stereopsis.

Materials and Methods

A total of 240 patients newly diagnosed with PD during this period were included in this study.

Data were collected through reviewing electronic medical records between March 1, 2021 and March 31, 2021. The personal information was kept confidential, and each individual was assigned anonymously. All data were entered into the electronic medical record at the time of the patients' initial diagnosis of PD. Two neurologists collected the data from medical records including clinical information such as histories, disease duration, symptoms, neurological examination results, scale scores, neurophysiological test results, cognitive function test (components of the Seoul Neuropsychological Screening Battery, and the Mini Mental State Examination) results and stereopsis test results. All tests and scores were conducted in the off state within one month of the patient’s diagnosis."

Comment 2: Some of the enrolled subjects may be diagnosed as atypical parkinsonism later. How long did the authors follow up after the diagnosis of PD? In particular, freezing of gait (FOG) is very rare in drug-naive PD patients, but can be seen in patients with atypical parkinsonism even from the early stage. In this study 7% of enrolled subjects had FOG, and it is better to check the data again.

Response 2: We agree that the symptom of freezing of gait (FOG) is very rare in early PD. The FOG we assessed was recorded as a binary item based on the patient’s self-report, which we believe may have lower reliability. We reviewed the medical records again in accordance with the reviewer's comments. But the follow-up period varied for each patient, and in some cases, it was difficult to track patients due to follow-up loss. It was not possible to definitively confirm whether the patient's final diagnosis had changed. Therefore, we decided to remove the FOG variable due to its low reliability.

Comment 3: The authors checked various symptoms, including constipation, anosmia, orthostatic dizziness, urinary incontinence, visual hallucination, easy falling and FOG. How did they define these symptoms? Any scale used to screen these symptoms (e.g. NMSS sub-items)?

Response 3: We thank the reviewer for the comment. Constipation, anosmia, orthostatic dizziness, urinary incontinence, visual hallucinations, easy falling, and FOG were recorded based on a simple yes or no format rather than using a validated questionnaire. Therefore, we replaced “constipation” with NMSS domain 6 (gastrointestinal tract), “orthostatic dizziness” with NMSS domain 1 (cardiovascular including falls), “urinary incontinence” with NMSS domain 7 (urinary), and “visual hallucinations” with NMSS domain 4 (perceptual problems/hallucinations) subscales. “Anosmia”, “FOG”, and “easy falling” were removed from the analysis. 

Table 1.

Comment 4: In terms of table 2, how about using sub-score of UPDRS part 3 (e.g. sub-score for tremor, rigidity, bradykinesia and axial symptoms)? This would be more easy for the readers.

Response 4: We thank the reviewer for the thoughtful comment. We have revised the table in accordance with the reviewer's suggestions. 

Table 2. 

Comment 5: Results without statistical significance like the comparison of GAITRITE data can be moved to supplement.

Response 5: We agree with the reviewer’s suggestion to move Table 3 to Supplement table 1.

Supporting information

S1 Table. 

Comment 6: Did the authors check RBD or dysautonomia (not just symptoms but autonomic function test or orthostatic hypotension)? Considering these symptoms can be markers to differentiate brain-first vs. body-first subtype, analysis with these variables can be also interesting.

Response 6: We thank the reviewer for the comment. Since we do not perform polysomnography as a routine test, we only have symptom history for RBD. However, we have results from the tilt table test, which inserted into the table 1.

Table 1. 

Thank you for your kind and helpful comments.

Response to Reviewer 2

Comments to the Author

This study is original and relevant in exploring the relationship between postural instability, gait disturbances, and stereopsis abnormalities in patients with Parkinson's disease. This topic has been little investigated to date. The results presented are interesting, suggesting a possible association between the PIGD subtype and visual dysfunction, which may open new perspectives on the impact of visual abnormalities on the disease's motor symptoms. However, despite the significant contribution, some aspects could be improved to strengthen the conclusions and increase the robustness of the findings. 

Thank you for reviewing our manuscript. Our answers to your queries are as follows:

Comment 1: The most critical aspect is the absence of significant differences in the objective gait parameters between the groups. Can the relationship between gait/postural alterations and stereopsis be sustained based on scores in some items of UPDRS-section III?

Response 1: We thank the reviewer for the thoughtful comment. We agree that the lack of significant differences in objective gait parameters represents a major limitation of this study. As mentioned in the discussion, the GaitRite system has limitations in measuring certain kinetics and kinematics parameters such as upper extremity motion and center of pressure. It is possible that subtle differences not captured by this equipment may be reflected in the UPDRS scores, which are based on human assessment. Therefore, considering the value of examinations conducted by trained experts, we consider that conclusions based on UPDRS-section III items remain valid. 

Discussion

"Second, additional gait analysis using GaitRite did not reveal any differences between the two groups. GaitRite, which we used to analyze gait quantitatively, can assess spatiotemporal parameters; but GaitRite is limited in its ability to assess certain kinetics and kinematics parameters such as upper extremity motion and center of pressure. It is possible that subtle differences not captured by this equipment may be reflected in the UPDRS scores, which are based on human assessment."

Comment 2: (Introduction) The introduction situates the study's relevance in the context of PD, mentioning stereopsis as an important non-motor symptom. However, the article could delve deeper into how visual dysfunction, specifically stereopsis, contributes to motor disturbances in PD. This more comprehensive exploration is crucial for providing a solid justification for the study and understanding the full impact of visual abnormalities on the disease's motor symptoms.

Response 2: We appreciate the insightful comment provided by the reviewer. First, we would like to apologize for any confusion caused by ambiguous expressions in the introduction. Our study results indicated that patients with higher PIGD scores had a higher risk of abnormal stereopsis. Since stereopsis is one of the sensory input information, it is believed that this information could also affect gait disturbance and posture, potentially creating a vicious cycle. Additionally, both the PIGD score and abnormal stereopsis may be associated with cholinergic nerve degeneration. We clarified this point and revised the introduction.

Introduction

"Also, patients with abnormal stereopsis have worse bradykinesia compared to those with normal stereopsis [7]. It has been observed that the group of PD with freezing of gait demonstrated worse stereoacuity compared to the group without this symptom [8]. However, researchers have not yet to elucidated the specific pathways as to why abnormal stereopsis occurs in PD patients, or how it is related to motor symptoms.

The pedunculopontine nucleus (PPN) plays a crucial role in gait disturbance and posture in PD. It is also an area that affects freezing of gait, a disabling symptom uniquely found in PD [9]. This region consists of neurons projecting both ascending and descending between the cortex, thalamus, basal ganglia, cerebellum, and spinal cord, performing various functions [10]. Among these, PPN cholinergic projections have an extended effect on midbrain dopaminergic systems by innervating neurons in the substantia nigra pars compacta and ventral tegmental area [10]. By modulating dopamine release, it can also influence striatal input from the cortex and thalamus [10].

The neurotransmitter balance in peripheral organs, such as the retina, also affects stereopsis. The cholinergic system and dopaminergic deficits in the retina are involved in interocular interaction and mixed percepts by regulating the balance of excitatory and inhibitory signals in the binocular visual cortex [11]. Abnormal stereopsis can be affected by decreased visual acuity of one eye and decreased color perception [3, 7]. A dopamine deficit in the retina and other structures of the peripheral visual system can impair visual function, including stereopsis [12, 13]."

Comment 3: (Methods) The adopted test to detect stereopsis is appropriate, but the authors do not mention its sensitivity and specificity in patients with PD. Please consider including this information to improve the interpretation of the results.

Response 3: We thank the reviewer for the excellent comment. We do not have the raw data from the stereopsis tests conducted on the control group, so it would be difficult to present sensitivity and specificity. However, we will reference previous studies conducted in the same lab [reference 6 and 14] and other relevant research [reference 17] in the manuscript. 

Materials & Methods

"Normal stereoscopic vision stereopsis was characterized as having an arc measurement of 60 seconds or less in the Titmus fly test [6, 14, 17]."

Comment 4: (Results and Discussion) The results show differences between the groups regarding postural instability and gait according to UPDRS section III. However, the objective gait analysis did not show significant differences between the groups, even in parameters closely related to postural and gait alterations such as single and double support time and CV of step and stride length. Are the scores in UPDRS items used to classify the PIGD subtype correlated with these gait parameters? In the absence of differences in objective gait measures, further explanation is needed for the association between PIGD and abnormal stereopsis. 

Response 4: We thank the reviewer for the excellent comment. We have added a table showing the correlation between the parameters measured in the gait analysis and the PIGD score (Table 3).

Table 3. The correlations between the PIGD score and parameters of gait analysis

Comment 5: (Results and Discussion) The study found a significant link between the PIGD subtype and abnormal stereopsis. However, based on postural control theories, the discussion regarding the connection between the PIGD subtype and stereopsis abnormalities must be deeper. The authors also suggested that cholinergic denervation in PD may be associated with abnormal stereopsis, but the explanation is unclear and does not effectively connect the study results to this theory. 

Response 5: We agree with the reviewer’s comment. We tried a more detailed discussion regarding cholinergic denervation. The impairment of posture that may occur due to abnormal multisensory input caused by abnormal stereopsis could induce a vicious cycle, potentially further exacerbating posture and gait.

Discussion

"Physiologically, cholinergic denervation is a major cause of PIGD features in PD [20]. The PPN, which plays a key role in posture and gait stability [21], is part of the PPN-laterodorsal tegmental complex and sends cholinergic projections to the thalamus [20]. This area is known to play an active role in sensorimotor control [21, 22]. There is a possibility that these projections are associated with abnormal stereopsis. Also, pPrevious study hasve also indicated that the cholinergic system is involved in stereopsis abnormalities [11]. The cholinergic system regulates the excitatory and inhibitory balance in the binocular visual cortex, and cholinergic enhancement in the visual cortex (V1) decreases inhibitory drive. This increases interocular interaction and mixed percepts [11]. Therefore, cholinergic denervation in PD may not only increase the risk of postural instability and gait disturbance but also increase the risk of stereopsis abnormalities. Additionally, abnormal stereopsis may lead to a qualitative degradation of visual information, potentially interfering with feedback and adaptation processes. This can further create a vicious cycle by negatively impacting posture and gait."

Comment 6: (Study limitations) Please make sure to include the unique cohort as one of the limitations to consider.

Response 6: Thanks for the reviewer’s comment. In response to the reviewer's comments, we have revised the discussion.

Discussion

"First, its retrospective study design and its being conducted at a single tertiary hospital are limitations. Additionally, it is possible that the enrolled patients constituted a unique cohort. Therefore, the generalizability of our results may be limited."

Thank you for your kind and helpful comments.

Response to Reviewer 3

Comments to the Author

Thank you for reviewing of our manuscript. Our answers to your queries are as follows:

Comment 1: The title is informative; however, it is long. The authors should consider shortening the title, and avoiding using the term retrospective because it is no longer recommended. Besides, this is a cross-sectional study, no matter if it uses data from a databank or if the authors seek the data prospectively.

Response 1: We agree with the reviewer’s thoughtful comment. We changed the title to “Postural instability and gait disturbance are associated with abnormal stereopsis in Parkinson's disease.”

Title

"Postural instability and gait disturbance are associated with abnormal stereopsis in Parkinson's disease"

Comment 2: (Abstract) The text should be shortened without losing information. The authors should insert the setting and the study design.

Response 2: Thank you for providing the insight. We added “designed a cross-sectional study and” in the Methods section of abstract.

Abstract - Metho

---

## [Decision Letter · Decision Letter 1]

24 Nov 2024

PONE-D-24-29202R1Postural instability and gait disturbance are associated with abnormal stereopsis in Parkinson's diseasePLOS ONE

Dear Dr. Koh,

Thank you for submitting your manuscript to PLOS ONE. After careful consideration, we feel that it has merit but does not fully meet PLOS ONE’s publication criteria as it currently stands. Therefore, we invite you to submit a revised version of the manuscript that addresses the points raised during the review process.

We look forward to receiving your revised manuscript.

Kind regards,

Keisuke Suzuki, MD, PhD

Academic Editor

PLOS ONE

Additional Editor Comments:

One of the reviewers expressed major criticisms and comments regarding the revised manuscript, and the authors need to address these appropriately.

Reviewers' comments:

Reviewer's Responses to Questions

**Comments to the Author**

1. If the authors have adequately addressed your comments raised in a previous round of review and you feel that this manuscript is now acceptable for publication, you may indicate that here to bypass the “Comments to the Author” section, enter your conflict of interest statement in the “Confidential to Editor” section, and submit your "Accept" recommendation.

Reviewer #1: All comments have been addressed

Reviewer #2: (No Response)

2. Is the manuscript technically sound, and do the data support the conclusions?

Reviewer #1: Yes

Reviewer #2: No

3. Has the statistical analysis been performed appropriately and rigorously? 

Reviewer #1: Yes

Reviewer #2: No

4. Have the authors made all data underlying the findings in their manuscript fully available?

Reviewer #1: Yes

Reviewer #2: Yes

5. Is the manuscript presented in an intelligible fashion and written in standard English?

Reviewer #1: Yes

Reviewer #2: Yes

6. Review Comments to the Author

Reviewer #1: (No Response)

Reviewer #2: After a detailed analysis of the modifications made in the manuscript and the authors' responses to the reviewers' comments, the following major criticisms and observations are noted:

1. Experience with Parkinson’s Disease Terminology: While the authors made an effort to address the feedback provided, their responses reveal a certain inexperience with PD-specific terminology and clinical nuances. This lack of precision could lead to misinterpretations about the patient profile and, consequently, about the study’s findings, compromising clarity and reliability.

2. Insufficient Justification for Inclusion and Exclusion Criteria: The choice to include patients who had recently initiated treatment, even within a month, requires a more robust methodological justification. The lack of in-depth discussion on this potential interference with the primary outcome limits the study’s interpretative and methodological consistency.

3. Weak Argument for Relying Solely on Clinical Assessment via UPDRS: While the authors rely on clinical assessment via the UPDRS to support their conclusions despite the lack of significant differences in objective gait assessment, the assertion of conducting evaluations by "experts" may not be wholly convincing, especially in light of some conceptual and terminological challenges noted in other areas of the manuscript.

4. Lack of Exploration of Differences in Objective Gait Assessment and Clinical Evaluation: A significant inconsistency in the manuscript concerns the authors' contradictory arguments regarding the dissociation between the GaitRite and UPDRS III results. On the one hand, the authors argue that the GaitRite system has limitations and may not fully capture subtle aspects of gait and postural instability in PD patients, which could explain why it did not yield significant differences between groups. However, they also assert that the GaitRite results correlate with the UPDRS scores, suggesting that they capture relevant aspects of gait impairment. This contradictory stance weakens the credibility of the interpretation and leaves readers uncertain about the reliability of the GaitRite data. Another critical aspect that requires further discussion in the manuscript is the finding related to the single support measure, which is a key metric in assessing postural instability. The authors reported a correlation between single support time and the UPDRS for one leg but not for the other. This asymmetry raises important questions that the authors did not address, particularly regarding the potential role of early-stage PD asymmetry in influencing the observed results.

7. PLOS authors have the option to publish the peer review history of their article (what does this mean?). If published, this will include your full peer review and any attached files.

Reviewer #1: No

Reviewer #2: **Yes: **Maria Elisa Pimentel Piemonte

---

## [Author Response · Author response to Decision Letter 1]

6 Jan 2025

Response to Reviewer 2

Comments to the Author

After a detailed analysis of the modifications made in the manuscript and the authors' responses to the reviewers' comments, the following major criticisms and observations are noted:

Thank you for your second review of our paper. We have answered each of your points below.

Comment 1: Experience with Parkinson’s Disease Terminology: While the authors made an effort to address the feedback provided, their responses reveal a certain inexperience with PD-specific terminology and clinical nuances. This lack of precision could lead to misinterpretations about the patient’s profile and, consequently, about the study’s findings, compromising clarity and reliability.

Response 1: We apologize for any terminological mistakes or parts that may have caused confusion in the first draft. In preparing this revised version, we have made every effort to use precise and clear terminology. 

Comment 2: Insufficient Justification for Inclusion and Exclusion Criteria: The choice to include patients who had recently initiated treatment, even within a month, requires a more robust methodological justification. The lack of in-depth discussion on this potential interference with the primary outcome limits the study’s interpretative and methodological consistency.

Response 2: Thank you for your thoughtful comments. The one-month limitation was established to mitigate potential confounding factors, particularly medication effects, given our study's focus on early disease states. This temporal limitation, however, prevents our findings from being representative of the entire Parkinson's disease population and, as the reviewer noted, may limit the generalizability of our results. We have addressed these concerns in the discussion section. 

"Discussion

First, its retrospective study design and its being conducted at a single tertiary hospital are limitations. Additionally, it is possible that the enrolled patients constituted a unique cohort. We specified that the tests were conducted within one month of clinical diagnosis to minimize the influence of medication and maintain uniformity in the early stages of the disease. However, this approach limits generalization to the entire PD population, and this limitation needs to be further explored through additional research in future studies."

Comment 3: Weak Argument for Relying Solely on Clinical Assessment via UPDRS: While the authors rely on clinical assessment via the UPDRS to support their conclusions despite the lack of significant differences in objective gait assessment, the assertion of conducting evaluations by "experts" may not be wholly convincing, especially in light of some conceptual and terminological challenges noted in other areas of the manuscript.

Response 3: We thank the reviewer for the comment. Our clinical examinations were performed by an expert, and GaitRite data were presented to ensure objectivity. The clinical examination can be considered reliable as the GaitRite data showed correlation with the UPDRS PIGD sub score. While previous studies have demonstrated that gait analysis can detect Parkinson's disease in early-stage patients, clinical assessment by trained professionals cannot be fully substituted, despite significant technological advances. Furthermore, there were methodological limitations in our gait analysis. For instance, we were unable to measure upper extremity movements, which are often observable in early-stage patients. There is a possibility that stereopsis abnormalities may be more closely associated with upper body rather than other body parts. 

"Discussion

Second, additional gait analysis using GaitRite did not reveal any differences between the two groups. Recent studies have demonstrated that combining mat with wearable devices or 3D cameras enables the detection of subtle movements in early-stage patients, which can be valuable for diagnostic purposes [30, 31]. However, the equipment in our hospital, which consisted of only a mat and an analytical computer, was limited in its ability to assess certain kinetics and kinematics parameters such as upper extremity motion and center of pressure. It is possible that differences not captured by this equipment may be reflected in the UPDRS scores, which are based on human assessment. It is not impossible for spatiotemporal parameters to detect gait abnormalities in early PD; however, the parameters we measured in our gait analysis may have been insufficient to establish a connection with the risk of abnormal stereopsis. Thus, further studies utilizing gait analysis with integrated upper extremity movement assessment are necessary to validate these assumptions."

Comment 4: Lack of Exploration of Differences in Objective Gait Assessment and Clinical Evaluation: A significant inconsistency in the manuscript concerns the authors' contradictory arguments regarding the dissociation between the GaitRite and UPDRS III results. On the one hand, the authors argue that the GaitRite system has limitations and may not fully capture subtle aspects of gait and postural instability in PD patients, which could explain why it did not yield significant differences between groups. However, they also assert that the GaitRite results correlate with the UPDRS scores, suggesting that they capture relevant aspects of gait impairment. This contradictory stance weakens the credibility of the interpretation and leaves readers uncertain about the reliability of the GaitRite data. Another critical aspect that requires further discussion in the manuscript is the finding related to the single support measure, which is a key metric in assessing postural instability. The authors reported a correlation between single support time and the UPDRS for one leg but not for the other. This asymmetry raises important questions that the authors did not address, particularly regarding the potential role of early-stage PD asymmetry in influencing the observed results.

Response 4: We thank the reviewer for the meticulous review. Studies using wearable devices or 3D cameras have demonstrated the potential to capture subtle movements in early-stage patients, making them valuable for diagnosis. However, our equipment is limited to checking spatiotemporal parameters based solely on foot trajectories. It lacks the capability to assess upper limb movements, such as abnormal arm swings, which are commonly observed gait abnormalities in early-stage PD. This limitation may account for the differences between clinical examination and gait analysis results. Regarding the asymmetric differences in single support time shown in Table 3, since there was no comparison with a control group, we cannot determine whether this represents an abnormality detectable only by GaitRite and not by clinical observation, or if it is a measurement error of the GaitRite system. 

"Discussion

Second, additional gait analysis using GaitRite did not reveal any differences between the two groups. Recent studies have demonstrated that combining mat with wearable devices or 3D cameras enables the detection of subtle movements in early-stage patients, which can be valuable for diagnostic purposes [30, 31]. However, the equipment in our hospital, which consisted of only a mat and an analytical computer, was limited in its ability to assess certain kinetics and kinematics parameters such as upper extremity motion and center of pressure. It is possible that differences not captured by this equipment may be reflected in the UPDRS scores, which are based on human assessment. It is not impossible for spatiotemporal parameters to detect gait abnormalities in early PD; however, the parameters we measured in our gait analysis may have been insufficient to establish a connection with the risk of abnormal stereopsis. Thus, further studies utilizing gait analysis with integrated upper extremity movement assessment are necessary to validate these assumptions."

---

## [Editor Report · Decision Letter 2]

8 Jan 2025

Postural instability and gait disturbance are associated with abnormal stereopsis in Parkinson's disease

PONE-D-24-29202R2

Dear Dr. Koh,

We’re pleased to inform you that your manuscript has been judged scientifically suitable for publication and will be formally accepted for publication once it meets all outstanding technical requirements.

Kind regards,

Keisuke Suzuki, MD, PhD

Academic Editor

PLOS ONE
---

## [Editor Report · Acceptance letter]

10 Jan 2025

PONE-D-24-29202R2 

PLOS ONE

Dear Dr. Koh, 

I'm pleased to inform you that your manuscript has been deemed suitable for publication in PLOS ONE. Congratulations! Your manuscript is now being handed over to our production team.

Kind regards, 

on behalf of

Dr. Keisuke Suzuki 

Academic Editor

PLOS ONE